# MAPKKKs in Plants: Multidimensional Regulators of Plant Growth and Stress Responses

**DOI:** 10.3390/ijms24044117

**Published:** 2023-02-18

**Authors:** Chen Xie, Liu Yang, Yingping Gai

**Affiliations:** State Key Laboratory of Crop Biology, College of Life Sciences, Shandong Agricultural University, Tai’an 271018, China

**Keywords:** MAPKKK, abiotic stress, biotic stress, plant growth

## Abstract

Mitogen-activated protein kinase kinase kinase (MAPKKK, MAP3K) is located upstream of the mitogen-activated protein kinase (MAPK) cascade pathway and is responsible for receiving and transmitting external signals to the downstream MAPKKs. Although a large number of MAP3K genes play important roles in plant growth and development, and response to abiotic and biotic stresses, only a few members’ functions and cascade signaling pathways have been clarified, and the downstream MAPKKs and MAPKs of most MAP3Ks are still unknown. As more and more signaling pathways are discovered, the function and regulatory mechanism of MAP3K genes will become clearer. In this paper, the MAP3K genes in plants were classified and the members and basic characteristics of each subfamily of MAP3K were briefly described. Moreover, the roles of plant MAP3Ks in regulating plant growth and development and stress (abiotic and biotic) responses are described in detail. In addition, the roles of MAP3Ks involved in plant hormones signal transduction pathway were briefly introduced, and the future research focus was prospected.

## 1. Introduction

Plants are constantly confronted with complex environmental factors during their growth and development. In order to make living cells respond and adapt to the external environment, so as to achieve the healthy growth of the plant, the changes in the extracellular environment must be transmitted from the outside to the inside of the cell in a specific way, and finally to the nucleus where gene expression changes may occur [1,2]. Plants have evolved a variety of environmental response mechanisms to minimize damage by activating multiple signal transduction pathways. Activation and deactivation of enzymes by phosphorylation/dephosphorylation of kinases and phosphorylases allow rapid and specific signal transduction, as well as amplification of external stimuli [1,3,4,5,6,7]. One particular signal transduction mechanism, the mitogen-activated protein kinase (MAPK) cascade pathway, plays an important role in many plants. The classic MAPK pathway consists of three core members, namely MAPK, MAPK kinase (MAPKK), and MAPKK kinase (MAPKKK, MAP3K). MAP3Ks phosphorylate downstream MAPKKs, which in turn phosphorylates the MAPKs, and transmit the exogenous signal step by step downstream to regulate a variety of biological events [5,8,9,10]. A large number of studies have shown that the MAPK cascade pathway is associated with the essential development process of the plant growth cycle, and it can respond biologically to stimuli when plants are subjected to stress, including biotic stress and abiotic stress, to ensure the survival of plants [8,10,11]. The MAPK cascade signaling pathway in the plant has three main features. The first is efficiency. Stimulating signals can be transmitted downward through the process of MAPK cascade pathway phosphorylation step by step to achieve a signal amplification effect, thus triggering a series of responses in cells [12]. Moreover, the interactions among the three members of the MAPK cascade are specific in different cell activities, including cell division, differentiation, and programmed death, as well as various responses to stress. In addition, each MAPK cascade pathway functions independently, despite there being crosstalk between different pathways [4,12]. Furthermore, the MAPK cascade pathway is relatively conservative in the evolution of eukaryotes from yeast to higher animals. However, compared with a large number of MAPK pathways identified in yeast and animals, far fewer pathways are identified in plants. With the development of forward and reverse genetics, CLUSTERED REGULAR INTERSPACED SHORT PALINDROMIC REPEATS/Cas9 (CRISPR/Cas9) technology, and other technologies, the MAPK cascade pathways in the plant have been further studied in recent years.

As the upstream member of the MAPK cascade pathway, the MAP3K family has the most members and extensive functions. It is an important link in receiving and transmitting signals and deserves more attention [2,4,13]. This review will help the researchers who are interested in MAP3K understand how the MAP3K signal pathway regulates stress signals and development signals, so as to design breeding strategies to improve crop resistance to stress and disease and improve grain yield [3].

## 2. Classification and Characteristics of Plant MAP3Ks

In 1993, Stafstrom and Duerr found MAPK family proteins MAPK D5 kinase and EXTRALLULAR REGULATED PROTEIN KINASES 1 (ERK1) for the first time in *Glycine max* and *Medicago sativa*, respectively [14,15]. Since then, research on the MAPK signaling pathway has been gradually carried out. As the first member of the phosphorylation cascade, MAP3K has more family members than MAPKK and MAPK, and the primary structures and functional domains of the members are quite different. The *NICOTIANA TABACUM PROTEIN KINASE 1* (*NPK1*) gene is the first gene encoding MAP3K isolated from plants [16]. As more and more plant genomes are being sequenced, more and more MAP3K genes have been identified in plants. According to the sequence characteristics of the catalytic domains of MAP3K, it can be further divided into three subfamilies, namely the MAPK ERK KINASE KINASE (MEKK) subfamily, the RAPIDLY ACCELERATED FIBROSARCOMA (Raf) subfamily, and the ZIPPER INTERACTING PROTEIN KINASE (ZIK) subfamily. Table 1 shows the characteristics of catalytic and regulatory domains of the three subfamilies of MAP3K. The catalytic domains of the MEKK subfamily are located in the N-terminal, C-terminal, or intermediate regions, and the conservative amino acid sequence of these regions is G(T/S)PX(F/Y/W)MAPEV. The catalytic domain of the Raf subfamily was located at the C-terminal, and the conservative amino acid sequence of the region was GTPEFMAPE(L/V)Y. As for the ZIK subfamily, most of the catalytic domains are located at the N-terminal, and the conservative amino acid sequence of the region is GTXX(W/Y)MAPE [17]. MAP3K acts as a kinase by recognizing and phosphorylating the S/T-X_3-5_-S/T sequence in the downstream MAPKK through the above catalytic and regulatory domains. Table 2 shows the classification and number of MAP3Ks in plant species that have been studied. There were 80 AtMAP3Ks found in *Arabidopsis thaliana*, 12 of which belong to the MEKK subfamily, amongst *MEKK1*, *ARABIDOPSIS NPK1-LIKE PROTEIN KINASE 1/2/3* (*ANP1/2/3*), and *AtMAP3K18* are more concerned [18,19,20]; there were 48 MAP3Ks belonging to Raf subfamily, *ENHANCED DISEASE RESISTANCE 1* (*EDR1*), and *CONSTITUTE TRIPLE RESPONSE 1* (*CTR1*) genes are typical representatives of them [21,22]. In *Oryza sativa*, there are 22 members of the MEKK subfamily, 43 members of the Raf subfamily, and ten members of the ZIK subfamily. The different classifications of MAP3K imply that the MAP3K family has various functions.

## 3. The Role of MAP3Ks in Plant Growth and Development

Many studies have shown that MAP3K regulates various physiological and biochemical processes throughout the growth cycle, including cell proliferation and differentiation, germ cell and embryonic development, branch formation and leaf development, root development, and so on [2,34,35,36]. The majority of these MAP3K genes are found in *Arabidopsis thaliana*, *Nicotiana tabacum*, *Oryza sativa* and *Solanum chacoense*. Mining MAP3K genes that regulate growth and development in other plant species will have a very broad prospect and space.

MEKK-like *NPK1* is a member of MAP3K and mainly regulates cell division and differentiation in tobacco, and its *A. thaliana* homologues are *ARABIDOPSIS NPK1-LIKE PROTEIN KINASE 1* (*ANP1*), *ANP2*, and *ANP3*. *NPK1* is the first MAP3K gene cloned and identified from plants [37] and is expressed in meristem and immature organs of tobacco and can be activated by tubulin NPK1-ACTIVATING KINESIN 1 (NACK1) in the M phase of the cell cycle. NPK1 regulates the formation of cell plates through NPK1-NQK1-NRK1 MAPK cascade pathways, and plays a key role in the process of cell differentiation and cell wall formation [38,39]. *NPK1*-silenced plant exhibits reduced cell size and an overall dwarf phenotype. Further, many stomata guard cells of *NPK1*-silenced plants were binucleate or had no nucleus, and cytokinesis was incomplete [37]. In addition, *ANP1*, *ANP2*, and *ANP3* can regulate microtubule organization and participate in *A. thaliana* cytokinesis process by activating ANP1/2/3-MKK6-MPK4 MAPK cascade pathway. In the presence of AtNACK1, MKK6/ANQ was activated by ANP1, and it was weakly activated when AtNACK1 and ANP3 were co-expressed. However, ANP3 cannot activate MKK6/ANQ. Homozygous *mkk6/anq2* and *mkk6/anq3* plants exhibited severe dwarfism with typical defects in cytokinesis [19,40]. 

The plant MAP3K subfamily, which regulates the development of germ cells and embryos, has only a few genes, including *AtMAP3Kε1*, *AtMAP3Kε2*, *Solanum chacoense*
*FERTILIZATION-RELATED KINASE 1* (*ScFRK1*), and *ScFRK2*. *AtMAP3Kε1* and *AtMAP3Kε2* play an important role in pollen development, but they are functionally redundant in *A. thaliana*. The expressions of MAP3Kε1 and MAP3Kε2 are cell-cycle regulated, and their transcription levels are the highest at the late stages of the cell cycle. In addition, most of the MAP3Kε1 proteins are localized to the plasma membrane of pollen cells. The double mutation of map3kε1/map3kε2 caused severe damage to the cytoplasmic membrane structure and triggered pollen cell death [41]. In *S. chacoense*, the expression patterns of *FRK1* and *FRK2* genes show obvious tissue specificity, especially in the fertilized ovule, both of which are involved in embryonic development. In addition, it is reported that the overexpression of *FRK2* will lead to a significant reduction in potato yield and delayed seed development [42].

MAP3K, which regulates leaf development and branching, is only found in *A. thaliana* and *O. sativa*. Miao et al. suggested that MEKK1 may play a direct role as a DNA binding protein and can bind to the promoter of WRKY DNA-BINDING PROTEIN 53 (WRKY53) gene, thus bypassing the downstream MAPK cascade to regulate the senescence process of *A. thaliana* leaves. MEKK1 was able to interact with the WRKY53 protein and phosphorylate it, thus increasing the DNA-binding activity of WRKY53 and the transcription of the reporter gene driven by a *WRKY53* promoter in vivo [18]. AtMAP3Kδ4, which is a member of Raf subfamily, is crucial to regulating plant growth and shoot branching. The transgenic *A. thaliana* plants overexpressing MAP3Kδ4 showed earlier bolting and more vigorous growth than wild-type plants [43]. In contrast, the transgenic plants overexpressing MAP3Kδ4 KINASE-NEGATIVE showed a highly branched phenotype; MAP3Kδ4 overexpression had no effect on branch number. MAP3Kδ4 transcripts were found in all tissues of *A. thaliana* seedlings and upregulated by auxin, suggesting that it functions in an auxin-dependent manner [43]. In addition, as the upstream of AtMAPKK2, the Raf-like gene *AtMAP3K17* can regulate the transformation of seedlings from vegetative growth to reproductive growth. Overexpression of *AtMAP3K17* will lead to premature bolting and seedling dwarfism [44]. Moreover, a recent study revealed that MAP3K4 (YODA) and HEAT SHOCK PROTEIN 90.1 (HSP90.1) are epistatic, and they may play a linear role in the same developmental pathway regulating plant stomata formation. HSP90 interacts with MAP3K4, affects its cellular polarization, and modulates the phosphorylation of downstream targets, such as MPK6 and SPEECHLESS [45]. In *O. sativa*, *INCREASED LEAF ANGLE 1* (*ILA1*) encodes a Raf-like MAP3K protein with serine/threonine kinase activity, which is predominantly residential in the nucleus and expressed in the vascular bundles of leaf lamina joints. ILA1 interacts with an unknown protein ILA1 INTERACTING PROTEINS (IIPs) and can phosphorylate IIP4 to regulate the mechanical strength of blade junctions [46]. 

*AtMEKK1* and *AtMAP3K4* play important roles in root development. In addition to regulating the senescence process of leaves, MEKK1 can also regulate root development by participating in the glutamate signaling pathway [47]. The *mekk1* mutant showed a phenotype of overall dwarf, shorter root hairs, and fewer lateral roots than the wild type of *A. thaliana* [47]. As the upstream of MAPK6, AtMAP3K4 plays an important role in root development and the regulation of microtubules during mitosis. In the *map3k4* mutant, the cell division of the main and lateral root is disordered, resulting in the termination of cytokinesis [48]. 

Microscopically, MAP3Ks affect cell division and differentiation; macroscopically, they affect the development of tissues, organs, and embryos. In addition to the above direct effects on growth and development, MAP3Ks also affects the circadian rhythm and some metabolic processes of plants, thus indirectly affects growth and development. NO LYSINE 1 (WNK1) is a member of the ZIK subfamily and can phosphorylate ARABIDOPSIS PSEUDO RESPONSE REGULATOR 3 (APRR3), a member of the TIMING OF CAB EXPRESSION 1 (TOC 1), in vitro, which may play an important role in the regulation of *A. thaliana* circadian rhythm [34,49]. In addition, AtMAP3K20 can directly interact with and phosphorylates AtMPK18 and is involved in the regulation of microtubule function in through such an atypical MAPK cascade pathway. AtMAP3K20 were ubiquitously expressed in many tissues, especially in pollen grains. Under normal growth conditions, compared with wild-type plants, the *map3k20* mutants have no obvious defects at any development stage. However, the seedling roots of *map3k20* mutants were significantly shorter than wild type plants when treated with oryzalin (microtubule-disrupting agent). Furthermore, *map3k20* mutants also showed defects in microtubule organization [50,51]. *MAP3K* genes in *A. thaliana* and *O. sativa* are involved in sugar metabolism and signaling pathways of gibberellin (GA), salicylic acid (SA), ethylene (ET), and jasmonic acid (JA). Sugar can not only provide energy for plant cells, but also as a signal molecule to regulate gene expression and affect multiple development and metabolic processes. *A. thaliana* SUGAR INSENSITIVE 8 (SIS8) regulates the development of sugar-resistant seedling in by interacting with UDP glucosyltransferase in the nucleus. The *sis8* mutant showed the phenotype of high concentration sugar tolerance [52]. MAP3K5 of *O. sativa* is widely expressed in various tissues and development stages, and regulates the size of cells by regulating the content of endogenous GA content, and affects plant height and yield of *O. sativa*. Overexpression of *OstMAP3K5* caused the increase in GA content, which in turn leads to enlarged cell size, and ultimately leads to an increase in plant height and yield [53]. SA significantly induced the expression of *AtRaf43*, *BnaMAP3K18*, and *BnaRaf28*, and inhibited the expression of *BnaZIK2*, *BnaRaf34*, and *BnaRaf36* at specific time points in *Brassica napus*, implying that these genes may be involved in SA signaling pathway [31,54]. *AtCTR1* and some MAP3K genes in *B. napus* are associated with ET signaling pathways. AtCTR1, a member of Raf MAP3K family, can interact with ET receptor ETHYLENE RECEPTORS 1 (ETR1) and block the activation of MAPKK9-MAPK3/MAPK6 pathway, and negatively regulate ET signal transduction by enhancing the degradation of transcription factor ETHYLENE-INSENSITIVE 3 (EIN3) [21]. In *B. napus*, the expression of *BnaZIK8* and *BnaRaf29* were induced by ET, while the expression of *BnaMAP3K18*, *BsnaZIK3*, and *BnaRaf36* was inhibited by ET. In addition, JA can induce the expression of *BnaRaf30*, but inhibit the expression of *BnaZIK3*, *BnaZIK4*, *BnaRaf36*, and *BnaRaf39* [31]. Table 3 and Figure 1 summarize the function of MAP3K in regulating plant growth and development.

## 4. Role of MAP3Ks in Plant Responses to Abiotic Stresses

Plants will be affected by various abiotic stresses (such as temperature stress, drought stress, high salt stress) during their growth [3,55,56,57,58,59]. Recent studies have shown that various external stresses can activate different plant MAP3K genes, and these genes play roles in stress-resistance through dependent or independent MAPK cascade signaling pathway, most of them are involved in response to drought, osmotic or salt stress, and only a few of them regulate temperature, oxidative or injury stress response.

### 4.1. MAP3K Regulating Drought and Osmotic Stress Response

Plant MAP3K genes associated with drought and osmotic stress tolerance belong to the Raf or MEKK subfamily. Most of them, including *MAP3K ABA AND ABIOTIC STRESS-RESPONSIVE RAF-LIKE KINASE* (*ARK*), *Raf6*, *Raf12*, *Raf35*, *Raf43* in *A. thaliana*, *DROUGHT-HYPERSENSITIVE MUTANT 1* (*DSM1*) in *O. sativa*, *MAP3K40* and *Raf19* in *Gossypium raimondii*, are classed into Raf subfamily. However, up to now, the MAPK cascade signaling pathway involved in the above genes has not been identified. MEKK genes related to drought and osmotic stress tolerance is only reported in *A. thaliana* and *G. raimondii*, and they are including *AtMAP3K1*, *AtMAP3K10*, *AtMAP3K14*, *AtMAP3K15*, *AtMAP3K16*, *AtMAP3K17*, *AtMAP3K18*, *AtMAP3K19*, and *GhMAP3K14*. Among them, *AtMAP3K17*, *AtMAP3K18*, and *GhMAP3K14* have been studied intensively and the complete MAPK cascade signaling pathway they participate in has also been identified.

Osmotic stress can activate ARK and SNF-RELATED SERINE/THREONINE-PROTEIN KINASE 2 (SnRK2) simultaneously, and ARK can interact with SnRK2.6/OPEN STOMATA 1 (OST1) and maintain its activity to trigger physiological reactions, such as stomatal closure, so as to protect plants under drought conditions. In *ark* mutants, the activity of SnRK2 decreased and the water loss rate of leaves increased [60]. In addition to *ARK*, there are also some Raf genes in *A. thaliana*, such as *Raf6*, *Raf12*, and *Raf35*, which are also up-regulated by ABA, indicating that these genes may participate in ABA signal pathway and drought resistance process [61]. Nasar virk et al. suggested that Raf-like gene *AtRaf43* is required for tolerance to osmotic stress, and knockout of *AtRaf43* attenuated the osmotic stress tolerance, but overexpression of *AtRaf43* did not affect the osmotic stress tolerance. In addition, *AtRaf43* is involved in ABA response, and knockout of *AtRaf43* resulted in increased ABA sensitivity, but overexpression of *AtRaf43* had no effect on ABA response [54]. *DSM1* is the only Raf gene involved in drought stress tolerance in *O. sativa*, and drought stress induce the expression of *DSM1*, which can participate in reactive oxygen species signal pathway by regulating the two peroxidase genes *POX22.3* and *POX8.1*. The expression level of POX22.3 and POX8.1 in *dsm1* mutants was significantly lower than that in wild-type plants, resulting in increased ROS damage, which led to the sensitivity of the mutant to drought stress, while overexpression of *DSM1* could improve the drought resistance in the seedling stage [62]. In cotton, *MAP3K40* can be induced by drought and act as a positive regulator in drought tolerance during seedling germination. MAPKK4 is known to be located downstream of MAP3K40, but the complete MAPK signaling pathway it involved needs to be further studied [63]. In contrast to MAP3K40, Raf19, as a negative regulator, inhibits the expression of ROS related antioxidant genes, resulting in the accumulation of ROS, thereby negatively regulating cotton drought resistance. Furthermore, it was shown that the expression of *GhRaf19* was inhibited by PEG and NaCl, and cold (4 °C) and H_2_O_2_ induced it expression [64].

In addition to Raf gene, many genes of MEKK subfamily play a key role in regulating plant drought stress tolerance. Some upstream regulators of MAP3K have been found. It was reported that *MAP3K18* regulates the drought resistance of *A. thaliana* through the MAP3K18-MAPKK3-MAPK1/2/7/13/14 cascade pathway, and the *MAP3K18* gene knockout mutant is hypersensitive to drought stress, and the overexpression of *MAP3K18* can significantly enhance the drought resistance of seedlings. In addition, MAP3K18 also regulated by other factors, the ubiquitin ligases RING DOMAIN LIGASE 1 (RGLG1) and RGLG2 can promote its degradation, and the transcription factor ABSCISIC ACID INSENSITIVE 4 (ABI4) can activate its expression [20,65]. It is also reported that drought stress can induce the accumulation of ABA, and the overexpression of *MAP3K18* will make the seedlings sensitive to ABA at the seedling growth stage. In contrast, the *MAP3K17* overexpression lines of another MEKK gene in *A. thaliana*, were insensitive to ABA, and the gene negatively regulated the ABA signaling pathway through a MAP3K17-MAPKK3-MAPK1/2/7/14 cascade pathway [44]. In addition to *AtMAP3K17* and *AtMAP3K18*, other MEKK genes such as *AtMAP3K1*, *AtMAP3K10*, *AtMAP3K14*, *AtMAP3K15*, *AtMAP3K16*, and *AtMAP3K19* were also up-regulated by ABA. Whether they participate in drought stress tolerance remains to be verified [61]. In cotton, only one MEKK gene *MAP3K14* was identified to participate in drought resistance process, and this gene positively regulates drought tolerance through the MAP3K14-MKK11-MPK31 cascade pathway. It was found that the single mutant of these three genes would lead to the reduction of drought tolerance of cotton seedlings [66]. 

### 4.2. MAP3K Regulating Salt Stress Response

MAP3K genes regulating salt tolerance were only identified in *A. thaliana* and *G. raimondii*, including *MEKK1* and *MAP3K20* in the MEKK subfamily and *AT6* and *Raf43* in the Raf subfamily of Arabidopsis, and *MAP3K40* and *Raf19* in the Raf subfamily of cotton [50,54,67,68].

*MAP3K20* was first found to be a regulator of salt stress response in *A. thaliana*, and it positively regulated salt tolerance through a cascade pathway consisting of MAP3K20, MKK3 and unknown MAPK. The *map3k20* mutants were sensitive to high salt and showed higher accumulation of reactive oxygen species under salt stress. In contrast, the *MAP3K20* overexpression lines showed tolerance to salt stress [50,51]. In addition, it was reported that the phenotypes of mutants and overexpression lines of the *MEKK1* gene were similar to those of *MAP3K20*. Salt stress can induce the expression of *MEKK1*, which can activate the expression of salt stress related genes through MEKK1-MKK2-MPK4/MPK6 cascade signaling pathway to positively regulate salt stress tolerance [67]. The *AtRaf43* mentioned above not only plays a role in drought stress tolerance, but also is necessary in salt stress tolerance. Knockout of *AtRaf43* attenuated the salt tolerance, while the overexpression of *AtRaf43* had no effect on salt s tolerance [54]. In addition to the above up-regulated genes induced by salt stress, the *AT6* repressed by salt stress was also found in *A. thaliana*. Gao and Xiang screened a mutant *at6* with high resistance to salt stress from the T-DNA insertion mutant library. Under salt stress, the expression of *AT6* was down regulated or closed, enabling the stress signal to be turned on and amplified, so that downstream effector genes could be rapidly expressed to cope with the stress [68]. In addition to responding to drought stress, cotton *MAP3K40* can also positively regulate salt stress tolerance, and it can be induced by salt stress, and its overexpression can improve plant germination rate under high salt stress [63]. In contrast to *MAP3K40*, it was reported that *Raf19* negatively regulates salt stress tolerance in cotton. Under salt stress, Raf19 inhibits the expression of ROS related antioxidant genes, leading to the accumulation of ROS [64].

### 4.3. MAP3K Regulating Temperature Stress Response

In plants, only a few genes of Arabidopsis and cotton have been identified as related to temperature stress tolerance, but their mechanisms have only been preliminarily explored. *MEKK1* not only acts as a positive regulator in salt stress tolerance, but also regulates cold stress tolerance through MEKK1-MKK2-MPK4/MPK6 cascade pathway in *A. thaliana*. Compared with wild-type or *mkk2* null plants, *MKK2*-overexpressing plants showed increased freezing tolerance [67]. In addition, *AtMAP3K20* and *GhMAP3K40* can be induced by low temperature stress, but the molecular mechanism has not been studied. GhRaf19 acts as a positive regulator to improve the ability of cotton to low temperature by activating the expression of ROS related antioxidant genes or enzymes under low temperature [63,64]. Cotton *MAP3K65*, belonging to Raf subfamily, can be induced by heat stress and various signal molecules (SA, JA and ET). Silence of *MAP3K65* can enhance the resistance of cotton to heat stress. In contrast, overexpression of *MAP3K65* enhances susceptibility to heat stress [69].

### 4.4. MAP3K Regulating Other Stress Response 

Studies of MAP3K related to other abiotic stresses in plants are few. *AtRaf43* is required for oxidative stress tolerance. Knockout of *Raf43* could weaken oxidative stress tolerance and increased the level of cellular damage under oxidative stress, but the overexpression of *Raf43* did not affect the oxidative stress tolerance of *A. thaliana* [54]. The research on *GhMAP3K40* only proved that the gene can be induced by oxidative stress and injury stress [63]. In *Medicago sativa*, it was found that the *OMTK1* gene which belongs to MEKK subfamily is activated by H_2_O_2_ rather than other hormones or stresses. After being activated by H_2_O_2_, OMTK1 directly interacts with a MAPK protein MMK3 and phosphorylates it and then forms a protein complex with it in vivo. OMTK1 plays a MAPK scaffolding role and functions in activation of H_2_O_2_-induced cell death in plants [70]. Table 4 and Figure 2 summarize the function of MAP3K in plant responses to abiotic stresses.

## 5. Role of MAP3K in Plant Responses to Biotic Stresses

Plants are not only subjected to various abiotic stresses during their growth and development, but also often threatened by biotic stresses such as pathogen invasion and insect feeding [71,72,73,74]. Studies have shown that plant receptors can recognize the conservative components of pathogen and produce complex immune responses to pathogen infection. The immune responses include cytoplasmic deposition, ROS burst, and activation of MAPKs [75,76,77]. There are many MAP3K genes of plants involved in abiotic stress response, and the reported related genes are mainly concentrated in Arabidopsis and rice. Most of them are involved in bacterial induced immune response, such as *Arabidopsis MAP3K3* and rice *ILA1*. A few *MAP3K* genes play a regulatory role both in fungal and bacterial tolerance, such as *AtMEKK1* and *AtEDR1*. Some genes have been reported to be involved in the process of virus tolerance, such as tobacco *NPK1*. Mining MAP3K of other plants involved in immune response will be one of the focuses of future research. In *A. thaliana*, *MEKK1* not only plays a role in plant growth and response to abiotic stress, but also in response to both bacterial and fungal pathogens. Asai et al. identified a complete plant MAPK cascade pathway (MEKK1, MKK4/MKK5, and MPK3/MPK6), which play a role in the downstream of the flagellin receptor FLAGELLIN-SENSING 2 (FLS2), a leucine-rich-repeat (LRR) receptor kinase. This research also proved that the activation of this MAPK cascade confers plant resistance to both bacterial and fungal pathogens, which showed that the signaling events initiated by different pathogens converge into a conserved MAPK cascade. When infected by the virulent bacterial pathogen *Pseudomonas syringae*, the transformed leaves expressing constitutively active MEKK1 (MEKK1a) displayed enhanced resistance to *P. syringae*. In addition, it was showed that when MEKK1a was transiently expressed in leaves, the development of soft-rot symptoms caused by the fungal pathogen Botrytis cinerea was also effectively suppressed [72]. Interestingly, another research showed that MEKK1, MKK1/MKK2, and MPK4 form a MAPK cascade pathway and negatively regulate plant immune responses. The autoimmunity phenotypes of *mekk1* mutants are caused by the activation of defense responses mediated by SUPPRESSOR OF *mkk1 mkk2 1* (SUMM1). SUMM1 encodes the MEKK2, which is directly targeted by MPK4 [78]. In addition, Raf gene *AtEDR1* negatively regulates defenses and directly modulates the MKK4/MKK5-MPK3/MPK6 cascade to fine-tune plant immunity. The *edr1* mutant plants showed resistance to *Golovinomyces cichoracearum* or *P. syringae* pv. *Tomato* (Pto) DC3000 and the EDR1 over-expressing plants showed enhanced susceptibility to powdery mildew, because there was more spores of *G. cichoracearum* produced in the plant than wild-type ones. Moreover, it was found that the *edr1* mutants have highly activated MPK3/MPK6 kinase activity and higher levels of MPK3/MPK6 proteins than wild type. In addition, it was showed that EDR1 negatively affects MKK4/MKK5 protein levels [22]. A recent study shows that MAP3Kδ-1 (MKD1), a new Raf-like MAP3K isolated from *A. thaliana*, is required for full immunity against bacterial and fungal infection, and its expression was induced by trichothecenes derived from Fusarium sporotrichioides. MKD1 interacted with MKK1 and MKK5 and phosphorylated them in vitro. The MKD1-MKK1/MKK5-MPK3/MPK6 cascade pathway is involved in the resistance against *P. syringae* pv. *Tomato* DC3000 by inhibiting SUMOylation of disease resistant proteins and phosphorylation of a mycotoxin detoxifying enzyme [79]. 

The MEKK gene *MAP3K3/MAP3K5* forms a kinase cascade with MKK4/MKK5 and MPK3/MPK6, and transmits defense signals downstream of various plant receptor kinases in *A. thaliana*. The loss of MAP3K3/MAP3K5 will reduce the activation of MPK3/MPK6 in response to different pathogen-associated molecular patterns (PAMPs), thus increasing the susceptibility to bacterial pathogens. These results indicate that there are antagonistic interactions between the developmental MAPK cascade pathway (MAP3K4-MKK4/5-MPK3/MPK6) and the immune signaling MAPK cascade pathway (MAP3K3/MAP3K5-MKK4/5-MPK3/MPK6) [80]. In addition to *A. thaliana*, there are MAP3K genes that regulate the immune process in other plants. *EDR1*, a homologous gene of *AtEDR1* in rice, can be induced by JA, SA, and endothelin, and negatively regulate bacterial resistance by activating ET synthesis pathway. *EDR1*-suppressing/knockout plants, which developed spontaneous lesions on the leaves, have enhanced resistance to bacterial blight disease caused by Xanthomonas oryzae pv. oryzae (Xoo). This resistance was associated with the accumulation of SA and JA [81]. Similar to *EDR1*, another Raf gene in rice, *ILA1*, also negatively regulates immune response. ILA1 mainly phosphorylated the threonine 34 at the N-terminal domain of MAPKK4, which possibly influenced the stability of MAPKK4. In addition, the N-terminal domain of ILA1 is required for its homodimer formation and its phosphorylation capacity. ILA1 can inhibit the downstream MAPKK4-MAPK cascade pathway by phosphorylating the MAPKK4 N-terminal domain, thus endowing rice with resistance to bacterial blight caused by *Xoo* [82]. Tomato MAP3Kε can activate downstream factors including MEK2, WOUND-INDUCED PROTEIN KINASE (WIPK), and SALICYLIC ACID-INDUCED PROTEIN KINASE (SIPK), thereby positively regulate cell death related to plant immunity and enhances plant immunity. Knock out of *MAP3Kε* impaired the resistance of tomato to Xanthomonas campestris and *P. syringae* strains. Similar to MAP3Kε, MAP3Kα also belongs to the MEKK subfamily, it is required for the hypersensitive response and resistance against *P. syringae*. Both MAP3Kα and MAP3Kε can regulate cell death by activating MEK2-WIPK/SIPK [83,84,85,86]. In cotton, *GhMAP3K65* may respond to pathogen infection by SA/JA/ET and ROS signaling pathways. Silencing of *GhMAP3K65* enhanced the resistance of cotton to *Ralstonia solanacearum*. In contrast, overexpression of GhMAP3K65 enhanced the susceptibility to *R. solanacearum* by inducing the production of pathogen-induced ROS (mainly H_2_O_2_) in transgenic *Nicotiana benthamiana* [69].

Only a few MAP3K genes are involved in the immune process against fungus or viruses. AtRaf36, a Raf-like protein kinase in Arabidopsis, was found to negatively regulate plant resistance to pathogen *Phytophthora parasitica* by targeting the downstream MKK2 kinase. Loss of *Raf36* enhances the resistance to *P. parasitica* and the resistance mediated by *Raf36* might be counteracted by *mkk2* mutation [87]. Tobacco NPK1 not only plays a role in the cytokinesis process, but also participates in the response to tobacco mosaic virus (TMV) resistance. Silencing expression of *NPK1* interferes with the function of disease-resistance genes *N*, *Bs2*, and *Rx*, but does not affect the resistance mediated by *Pto* and *Cf4*. *NPK1* plays a role in regulating the resistance responses mediated by *N*- and *Bs2*- and may play a role in one or more MAPK cascade pathways, regulating multiple cellular processes [37]. Table 5 and Figure 3 summarize the function of MAP3K in plant responses to biotic stresses.

## 6. Conclusions and Future Prospective

The MAP3K family is a plant regulator with complex functions. It regulates all stages and aspects of plant growth and development through phosphorylation of downstream MAPKK, and plays an important role in plant biotic and abiotic stress tolerance. We found that MAPK cascades are complex and diverse, which is mainly reflected in that the same MAP3K can form different cascade pathways with different MAPKKs and MAPKs, and different MAP3Ks can form different cascade pathways with the same downstream MAPKKs or MAPKs, and different cascades interact and antagonize each other. In addition, a MAP3K can simultaneously regulate different stress responses or different aspects of growth and development, which is achieved through different cascade pathways. Furthermore, plant MAP3K can also interact with other downstream participants to play different biological functions in a way that does not depend on the MAPK cascade pathway. As this research moves forward, aiming to uncover the novel functions and regulatory mechanisms of MAP3Ks, in addition to those described here will likely be uncovered. Figure 4 shows a comprehensive model depicting the overall interplay of plant MEKKs in regulating key physiological processes with their downstream players.

Mining the key MAP3K genes in plants, especially in crops, and the complete MAPK cascade pathway they participate in will still be the key work in the future. A variety of biological methods such as virus induced gene silencing (VIGS), RNA interference (RNAi), and CAS9 technology have also been gradually applied to the study of MAP3K function and have achieved some encouraging results. Using gene editing technology to knock out or overexpress the key MAP3K in crops, to cultivate crops with better agronomic traits and stronger tolerance to environmental stress is also the focus of future work. In addition, as the upstream of the MAPK cascade pathway, the knockout of MAP3K, which regulates multiple biological activities may lead to a variety of traits in plants, which may be beneficial or harmful, which are worthy of in-depth study in the future. Furthermore, it is also valuable to search for regulatory factors upstream of MAP3K, such as transcription factors, protein kinases, lncRNA, etc., and revealing how these regulatory factors regulate growth and development and stress responses will help to increase our understanding of MAP3K complex functions in plants. 

## Figures and Tables

**Figure 1 ijms-24-04117-f001:**
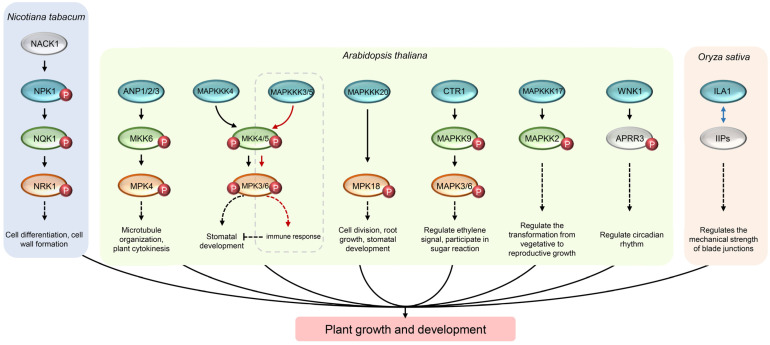
Mechanism of plant MAP3K regulating growth and development. The blue, green, and red ovals represent MAP3K, MAPKK, and MAPK respectively. White ovals represent proteins that do not belong to the MAPK cascade pathway. The red small circle with “P” represents phosphorylation and blue double-headed arrow represents interactions between proteins.

**Figure 2 ijms-24-04117-f002:**
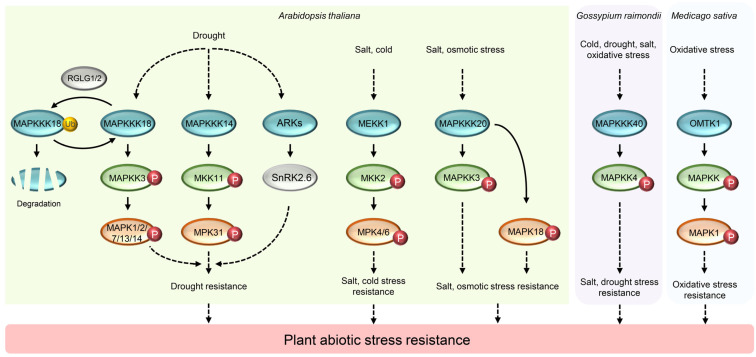
Mechanism of plant MAP3K regulating abiotic stress resistance. The blue, green, and red ovals represent MAP3K, MAPKK, and MAPK respectively. White ovals represent proteins that do not belong to the MAPK cascade pathway. The red small circle with “P” represents phosphorylation and the small yellow circle with “Ub” represents ubiquitination.

**Figure 3 ijms-24-04117-f003:**
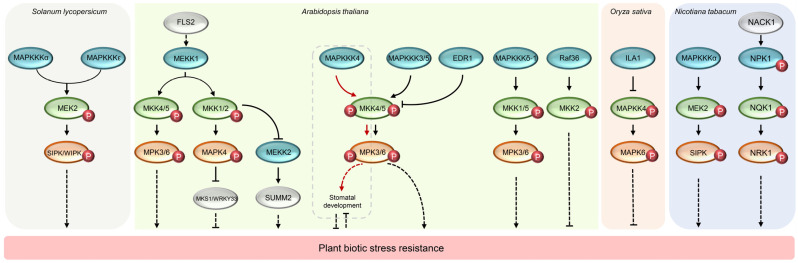
Mechanism of plant MAP3K regulating biotic stress resistance. The blue, green, and red ovals represent MAP3K, MAPKK, and MAPK respectively. White ovals represent proteins that do not belong to the MAPK cascade pathway. The red small circle with “P” represents phosphorylation.

**Figure 4 ijms-24-04117-f004:**
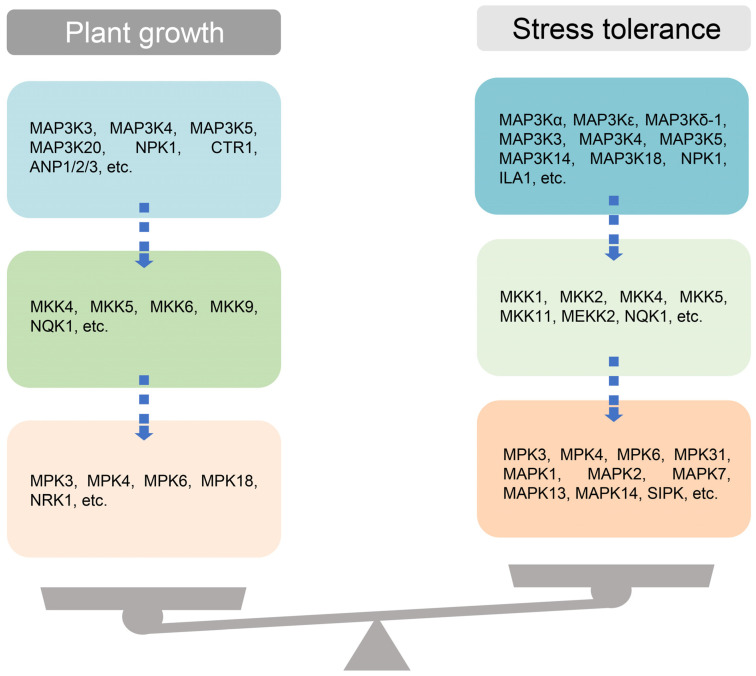
The working model of plant MAP3Ks. Plants form a dynamic balance between plant growth and stress tolerance through a series of MAPK cascade pathways composed of multiple MAP3Ks.

**Table 1 ijms-24-04117-t001:** Characteristics of three subfamilies of plant MAP3Ks.

Subfamily	Catalytic Domain	Regulatory Domain	Catalysed Region Sequence
MEKK	C-termianl, N-terminal, middle	ND	G (T/S) PX (F/Y/W) MAPEV
Raf	C-terminal	N-terminal	GTPEFMAPE (L/V) Y
ZIK	N-terminal	C-terminal	GTXX (W/Y) MAPE

Abbreviation: ND, not determined.

**Table 2 ijms-24-04117-t002:** Classification and quantity of plant MAP3Ks.

Species	MEKK	Raf	ZIK	Total	References
Apple (*Malus domestica*)	37	72	14	123	[23]
Cotton (*Gossypium raimondii*)	37	65	12	114	[24]
Pear (*Pyrus bretschneideri*)	57	31	20	108	[25]
Tomato (*Solanum lycopersicum*)	33	40	16	89	[26]
Arabidopsis (*Arabidopsis thaliana*)	12	48	20	80	[27]
Banana (*Musa nana*)	14	48	15	77	[28]
Rice (*Oryza sativa*)	22	43	10	75	[29]
Maize (*Zea mays*)	26	37	8	71	[30]
Rapeseed (*Brassica napus*)	18	39	9	66	[31]
Cucumber (*Cucumis sativus*)	18	31	10	59	[32]
Chinese jujube (*Ziziphus jujuba*)	15	41	0	56	[33]

**Table 3 ijms-24-04117-t003:** The MAP3Ks involved in plant growth and development.

Species	Gene	Subfamily	Function	Pathway
*Nicotiana tabacum*	*NPK1*	MEKK	Regulate cell division and differentiation and participate in cell wall formation	NPK1-NQK1-NRK1
*Arabidopsis thaliana*	*ANP1/2/3*	MEKK	Regulate microtubule tissue and cytokinesis	ANP1/2/3-MKK6-MPK4
*Arabidopsis thaliana*	*MEKK1*	MEKK	Involve in leaf senescence and root development	ND
*Arabidopsis thaliana*	*WNK1*	ZIK	Regulate day-night rhythm	WNK1-APRR3
*Arabidopsis thaliana*	*MAP3Kε1*/*ε2*	MEKK	Regulate pollen development	ND
*Arabidopsis thaliana*	*MAP3Kδ4*	Raf	Regulate both plant growth and shoot branching	ND
*Arabidopsis thaliana*	*MAP3K4*	MEKK	Regulate stomatal development	MAP3K4-MKK4/5-MPK3/6
*Arabidopsis thaliana*	*MAP3K20*	MEKK	Adjust the function of microtubule	MAP3K20-MPK18
*Arabidopsis thaliana*	*CTR1*	Raf	Regulate ethylene signal and participate in sugar reaction process	CTR1-MAPKK9-MAPK3/6
*Arabidopsis thaliana*	*MAP3K17*	Raf	Participate in the transformation from vegetative growth to reproductive growth	MAP3K17-MAPKK2
*Arabidopsis thaliana*	*SIS8*	Raf	As an anti-hyperglycemic regulator	ND
*Oryza sativa*	*ILA1*	Raf	Participate in the formation of mechanical organization at the blade joint	ND
*Oryza sativa*	*MAP3K5*	Raf	Regulate cell size	ND
*Solanum chacoense*	*FRK1/2*	MEKK	Participate in embryonic development	ND

Abbreviation: ND, not determined.

**Table 4 ijms-24-04117-t004:** The MAP3Ks involved in plant abiotic stress.

Species	Gene	Subfamily	Stress	Pathway
*Arabidopsis thaliana*	*MAP3K18*	MEKK	Drought	MAP3K18-MAPKK3-MAPK1/2/7/13/14
*Arabidopsis thaliana*	*MAP3K17*	MEKK	Drought	MAP3K17-MAPKK3-MAPK1/2/7/14
*Arabidopsis thaliana*	*ARK*	Raf	Osmotic	ND
*Arabidopsis thaliana*	*AT6*	Raf	Salt	ND
*Arabidopsis thaliana*	*MEKK1*	MEKK	Salt, cold	MEKK1-MKK2-MPK4/6
*Arabidopsis thaliana*	*MAP3K20*	MEKK	Salt, cold, osmotic	MAP3K20-MAPKK3
*Arabidopsis thaliana*	*Raf43*	Raf	Salt, drought, oxidative, osmotic	ND
*Medicago sativa*	*OMTK1*	MEKK	Oxidative	OMTK1-MMK3
*Oryza sativa*	*DSM1*	Raf	Drought, salt	ND
*Gossypium raimondii*	*MAP3K14*	MEKK	Drought	MAP3K14-MKK11-MPK31
*Gossypium raimondii*	*MAP3K40*	Raf	Cold, salt, drought, oxidative, damage, pathogen	MAP3K40-MAPKK4
*Gossypium raimondii*	*MAP3K65*	Raf	Heat	ND
*Gossypium raimondii*	*Raf19*	Raf	Salt, drought, cold	ND

Abbreviation: ND, not determined.

**Table 5 ijms-24-04117-t005:** The MAP3Ks involved in plant biotic stress.

Species	Gene	Subfamily	Stress	Pathway
*Arabidopsis thaliana*	*MEKK1*	MEKK	Both bacterial and fungal	MEKK1-MKK4/5-MPK3/6 and MEKK1-MKK1/2-MAPK4
*Arabidopsis thaliana*	*MAP3Kδ-1*	Raf	Both bacterial and fungal	MAP3Kδ-1-MKK1/5-MPK3/6
*Arabidopsis thaliana*	*EDR1*	Raf	Both bacterial and fungal	EDR1-MKK4/5-MPK3/6
*Arabidopsis thaliana*	*Raf36*	Raf	Only fungal	Raf36-MKK2
*Arabidopsis thaliana*	*MAP3K3/5*	MEKK	Only bacterial	MAP3K3/5-MKK4/5-MPK3/6
*Solanum lycopersicum*	*MAP3Kα*	MEKK	Only bacterial	MAP3Kα-MEK2-SIPK/WIPK
*Solanum lycopersicum*	*MAP3Kε*	MEKK	Only bacterial	MAP3Kε-MEK2-SIPK/WIPK
*Gossypium raimondii*	*MAP3K65*	Raf	Only bacterial	ND
*Oryza sativa*	*EDR1*	Raf	Only bacterial	ND
*Oryza sativa*	*ILA1*	Raf	Only bacterial	ILA1-MAPKK4-MAPK6
*Nicotiana tabacum*	*NPK1*	MEKK	Only viruses	NPK1-NQK1-NRK1

Abbreviation: ND, not determined.

## Data Availability

Not applicable.

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
