# Peer review of "MAPKKKs in Plants: Multidimensional Regulators of Plant Growth and Stress Responses"

_ijms, 2023, doi:10.3390/ijms24044117_

Round 1

Reviewer 1 Report

Overall this manuscript has substantial content but very casually written which needs thorough revision. Manuscript is full of gramatticall errors and very casually discussed. Across the manuscript only probable or identified pathways are mentioned, there is no any respond on biochemical or physiological level included, hence, authors are suggested to include all suggestions mentioned over the text. This manuscript also not clearly concluded or future perspectives mentioned, needs inclusion.

Author Response

Response to reviewer #1:

Q1: Manuscript is full of gramatticall errors and very casually discussed.

Response 1: We are very sorry for the grammatical error. Based on your suggestion, we have carefully revised the manuscript and the use of grammar have been checked and corrected by a native English speaker. We feel the revised manuscript is greatly improved as a result.

Q2: Across the manuscript only probable or identified pathways are mentioned, there is no any respond on biochemical or physiological level included, hence, authors are suggested to include all suggestions mentioned over the text.

Response 2: Thank you very much for your valuable suggestions. We have revised the manuscript and added a series of contents about the effects of MAP3K on plant at biochemistry or physiological level (page 4, line 139-140; page 5, line 147-149, 160-162, 169-170; page 6, line 202-203; page 9, line 292-293; page 10, line 360-361; page 13, line 423-429, 449-451).

Q3: This manuscript also not clearly concluded or future perspectives mentioned, needs inclusion.

Response 3: Based on your suggestion, the discussion sections have been reorganized and rewritten, we have added the clear conclusion and future perspectives in line 486-514.

Reviewer 2 Report

Given the role of MAP kinases in regulating various crucial physiological processes in plants, the current review manuscript detailing the role of MAPKKKs or MEKKs is of high interest. The authors have given a comprehensive description of the plant MEKK roles during different conditions, including stress responses. However, I have a few suggestions to improve the manuscript. 

1.  The authors have submitted a review article. Hence, no need to use the headings as results discussion etc. Rather they should use compelling conceptual headings under which they can present the different crucial information related to MEKKs.

2. Although MAPKKK is an accepted way of writing the name, I suggest the authors to use the shortened versions like MEKKs or MAP3Ks. They can use the MAPKKK at the first and then can switch to the alternative names. This is to improve the readability of the text.

3. The authors have presented individual models/scenarios of MEKKs signaling during different processes. I advise the authors to also include a comprehensive model depicting the overall interplay of plant MEKKs in regulating key physiological processes with their downstream players. 

4. At recent times, the targeted genome engineering has been in lime light. The authors are requested to include the application of such techniques (CRISPR, base, prime editing) in MEKKs to improve plant performances. Also, the MEKKs are the upstream players and can regulate multiple signaling or processes. So, will their editing affect more than one process in plant? The authors can also discuss this possibility.  

Author Response

Response to reviewer #2:

Q1: The authors have submitted a review article. Hence, no need to use the headings as results discussion etc. Rather they should use compelling conceptual headings under which they can present the different crucial information related to MEKKs.

Response 1: Thank you very much for your suggestions. We have deleted the headings as discussion results etc. and replaced it with the headings that can present the different crucial information related to MAP3K in line 116, 242, 253, 318, 346, 364, 387, 486.

Q2: Although MAPKKK is an accepted way of writing the name, I suggest the authors to use the shortened versions like MEKKs or MAP3Ks. They can use the MAPKKK at the first and then can switch to the alternative names. This is to improve the readability of the text.

Response 2: Based on your suggestion, we have checked the manuscript carefully, and the “MAPKKK” is used only at the first, other “MAPKKK” are replaced by “MAP3K” in the revised manuscript.

Q3: The authors have presented individual models/scenarios of MEKKs signaling during different processes. I advise the authors to also include a comprehensive model depicting the overall interplay of plant MEKKs in regulating key physiological processes with their downstream players.

Response 3: Based on your suggestion, we have added figure 4, which describes the overall pattern of plant MAP3K and its downstream players in growth and stress tolerance.

Q4: At recent times, the targeted genome engineering has been in lime light. The authors are requested to include the application of such techniques (CRISPR, base, prime editing) in MEKKs to improve plant performances. Also, the MEKKs are the upstream players and can regulate multiple signaling or processes. So, will their editing affect more than one process in plant? The authors can also discuss this possibility.

Response 4: Thank you very much for your suggestion. Yes, we agreed with you that the editing of MAP3K gene that regulates multiple biological activities will affect many aspects of plants. Based on your suggestion, we discussed the application of CRISPR in MAP3K in the revised manuscript. in line 503-513.
